# KMT2D/MLL2 inactivation is associated with recurrence in adult-type granulosa cell tumors of the ovary

R. Tyler Hillman[1,2], Joseph Celestino[1], Christopher Terranova[2], Hannah C. Beird[2], Curtis Gumbs[2], Latasha Little[2], Tri Nguyen[1], Rebecca Thornton[2], Samantha Tippen[2], Jianhua Zhang[2], Karen H. Lu[1], David M. Gershenson[1], Kunal Rai[2], Russell R. Broaddus[3] & P. Andrew Futreal[2]

Adult-type granulosa cell tumors of the ovary (aGCTs) are rare gynecologic malignancies that exhibit a high frequency of somatic FOXL2 c.C402G (p.Cys134Trp) mutation. Treatment of relapsed aGCT remains a significant clinical challenge. Here we show, using whole-exome and cancer gene panel sequencing of 79 aGCTs from two independent cohorts, that truncating mutation of the histone lysine methyltransferase gene KMT2D (also known as MLL2) is a recurrent somatic event in aGCT. Mono-allelic KMT2D-truncating mutations are more frequent in recurrent (10/44, 23%) compared with primary (1/35, 3%) aGCTs ($p = 0.02$, two-sided Fisher's exact test). IHC detects additional non-KMT2D-mutated aGCTs with loss of nuclear KMT2D expression, suggesting that non-genetic KMT2D inactivation may occur in this tumor type. These findings identify KMT2D inactivation as a novel driver event in aGCTs and suggest that mutation of this gene may increase the risk of disease recurrence.

[1] Department of Gynecologic Oncology and Reproductive Medicine, The University of Texas MD Anderson Cancer Center, Houston, TX 77030, USA. [2] Department of Genomic Medicine, The University of Texas MD Anderson Cancer Center, Houston, TX 77030, USA. [3] Department of Pathology, The University of Texas MD Anderson Cancer Center, Houston, TX 77030, USA. Correspondence and requests for materials should be addressed to P.A.F. (email: afutreal@mdanderson.org)

Granulosa cell tumors of the ovary (GCTs) are rare malignancies arising from the ovarian sex cord stroma that account for 2–5% of the approximately 239,000 ovarian cancers diagnosed each year worldwide[1]. Similar to their hormonally active cell type of origin, GCTs often produce estrogen, in turn leading to endometrial proliferation and associated symptomatic uterine bleeding[2]. As a result, 80–90% of GCTs are detected at an early stage, with surgery being curative in the majority of cases when tumor remains confined to the ovary[2]. GCT recurrence remains largely unpredictable, although this risk may be increased by factors such as initial stage[3,4], incomplete staging at the time of initial surgery[5], and intraoperative tumor rupture[6]. The optimal treatment of recurrent GCTs has not been established, but in current practice often includes a combination of surgery and cytotoxic chemotherapy[7].

GCTs are divided histologically into the more common adult-type (aGCTs), accounting for approximately 95% of GCTs, with the remainder being the less common juvenile subtype. Genetic investigation has identified the presence of *FOXL2* c.C402G (p.Cys134Trp) somatic mutation in 92–97% of aGCTs, although the presence of this hotspot mutation is not prognostic[8–10]. In addition, it was recently shown that activating *TERT* C228T promoter mutations are present in approximately 40% of recurrent aGCTs, a rate that is significantly higher than that observed in primary tumors[11,12].

In this study, we have used whole-exome and cancer gene panel sequencing to analyze 79 primary and recurrent aGCTs from two independent cohorts. In both cohorts we identify truncating mutation of the histone lysine methyltransferase gene *KMT2D* (also known as *MLL2*) as a recurrent somatic event in aGCT. We found that mono-allelic *KMT2D*-truncating mutations are significantly more common in recurrent compared to primary aGCTs. Moreover, immunohistochemistry (IHC) identified loss of KMT2D protein expression in some tumors without detectable *KMT2D*-truncating mutations, suggesting that non-genetic causes of *KMT2D* inactivation may also occur in this tumor type. Our findings demonstrate an association between *KMT2D* inactivation and aGCT relapse, providing new insight into the molecular pathogenesis of this rare disease.

## Results

**Mutational landscape of primary and recurrent aGCT.** To investigate the genomic landscape of primary and recurrent aGCT, we first identified an exploratory cohort comprised of cryopreserved aGCT surgical biopsies from 24 patients (primary, $n = 8$; recurrent, $n = 16$), with 20 having matched normal tissue or peripheral blood samples (Supplementary Data 1). We performed whole-exome sequencing (WES) on these tumor samples at 200× target coverage (mean 226; range 165–296) and also performed WES on the available normal samples (Supplementary Data 2). When examining only tumors for which a matched normal sample was analyzed, we detected 3443 (median 159; range, 87–449) somatic single-nucleotide variants (SNVs) and small insertions/deletions (indels). The frequency of somatic mutation (SNVs + indels) in recurrent aGCTs (median 2.1 mutations per megabase (Mb), interquartile range (IQR) 1.4–2.9) was higher than in primary aGCTs (median 1.2 mutations per Mb, IQR 0.7–1.7; $p = 0.01$, two-sided Wilcoxon's rank-sum test) (Fig. 1a). The time interval between diagnosis and sample collection as well as prior exposure to cytotoxic chemotherapy varied among recurrent tumors in this cohort (Fig. 1b).

We identified a somatic *FOXL2* c.C402G mutation[8] in 23 of 24 tumor samples (96%) in this exploratory cohort, a rate consistent with previous reports and supportive of the pathologic identification of these tumors as aGCTs (Fig. 1c)[8,10]. The only other somatic mutation identified in *FOXL2* was loss of the stop codon resulting from an 1129dupT indel. This indel was identified in two aGCTs in the exploratory cohort and is predicted to result in the addition of 157 amino acids to the C terminus of the 376 amino acid FOXL2 protein. We next used GenomeMuSiC2[13] to identify recurrent mutated genes in aGCTs with a matched normal sample. In addition to *FOXL2* ($q$ value $<10^{-13}$), we identified *KMT2D* ($q$ value $3.7 \times 10^{-7}$) as a significantly mutated gene in aGCT (Fig. 1c). All *KMT2D*-truncating mutations identified in the exploratory cohort were verified using deep amplicon sequencing (see Methods). When normalized to tumor cell fraction, the variant allele frequency (VAF) for four of the five *KMT2D* indels observed in the exploratory cohort fell between 0.29 and 0.36, with the normalized VAF of the remaining indel being 0.88 (see Supplementary Data 3).

Analysis of copy number alterations identified recurrent focal deletion of 21q22.3 (9/24 tumors, 38%; $q$ value $1.3 \times 10^{-5}$) and 3p29 (7/24 tumors, 29%; $q$ value $1.3 \times 10^{-5}$) in the exploratory cohort (Fig. 1d). Recurrent arm-level events previously observed in aGCTs including gain of 14q (11/24 tumors, 46%; $q$ value $<10^{-14}$) and loss of 22q (15/24 tumors, 63%; $q$ value $<10^{-14}$)[14] were also identified (Fig. 1e).

**KMT2D-truncating mutation in the KGN cell line.** The KGN cell line is a widely used model system for aGCT[15,16]. This cell line was originally derived from a recurrent metastatic aGCT and carries the *FOXL2* c.C402G hotspot mutation that is characteristic of this tumor type[10]. We performed deep amplicon sequencing of the *KMT2D* coding regions (Supplementary Data 4) in the KGN cell line and verified a truncating *KMT2D* mutation (c.4085delA; p.Asn1362fs) previously cataloged by the Cancer Cell Line Encyclopedia[17] and the COSMIC Cell Lines Project[18]. This truncating mutation was located prior to the enzymatic SET domain, similar to those truncating *KMT2D* mutations identified in our tumor cohorts.

**Cancer gene panel sequencing of validation cohort.** In order to further evaluate the association between *KMT2D*-truncating mutations and aGCT recurrence, we next performed a focused genomic analysis on an independent validation cohort comprised of 55 (primary, $n = 27$; recurrent, $n = 28$) formalin-fixed paraffin-embedded (FFPE) tumor samples and matched normal samples. To do this, we used a cancer gene hybrid capture platform previously validated for the analysis of FFPE tissue (Supplementary Data 5)[19]. In this validation cohort, a somatic *FOXL2* c.C402G mutation[8] was identified in 49 of 55 tumor samples (89%). Using stringent filtering criteria (see Methods), we identified six somatic *KMT2D*-truncating mutations in this validation cohort (6/55 tumors, 11%) including three frameshift indels, two nonsense mutations, and one splice site mutation (Table 1). Of the six *KMT2D* missense SNVs identified in the exploratory cohort, five were found in tumors without a matched normal sample and may in fact be rare SNPs. The remaining missense SNV (c.G15149T; p.Lys5050Arg) may be functionally deleterious since multiple missense mutations in this region of *KMT2D* have previously been shown to disrupt KMT2D methyltransferase activity in vitro[20].

**Association between KMT2D-truncating mutation and relapse.** When examining non-silent coding mutations in the exploratory cohort ($n = 24$ tumors) and validation cohort ($n = 55$ tumors) together, we identified a total of 11 *KMT2D* mutations predicted to result in reading frame truncation prior to the enzymatic SET domain (Fig. 2a). We found that such lesions were enriched in recurrent tumors across two independent cohorts (Fig. 2b). For

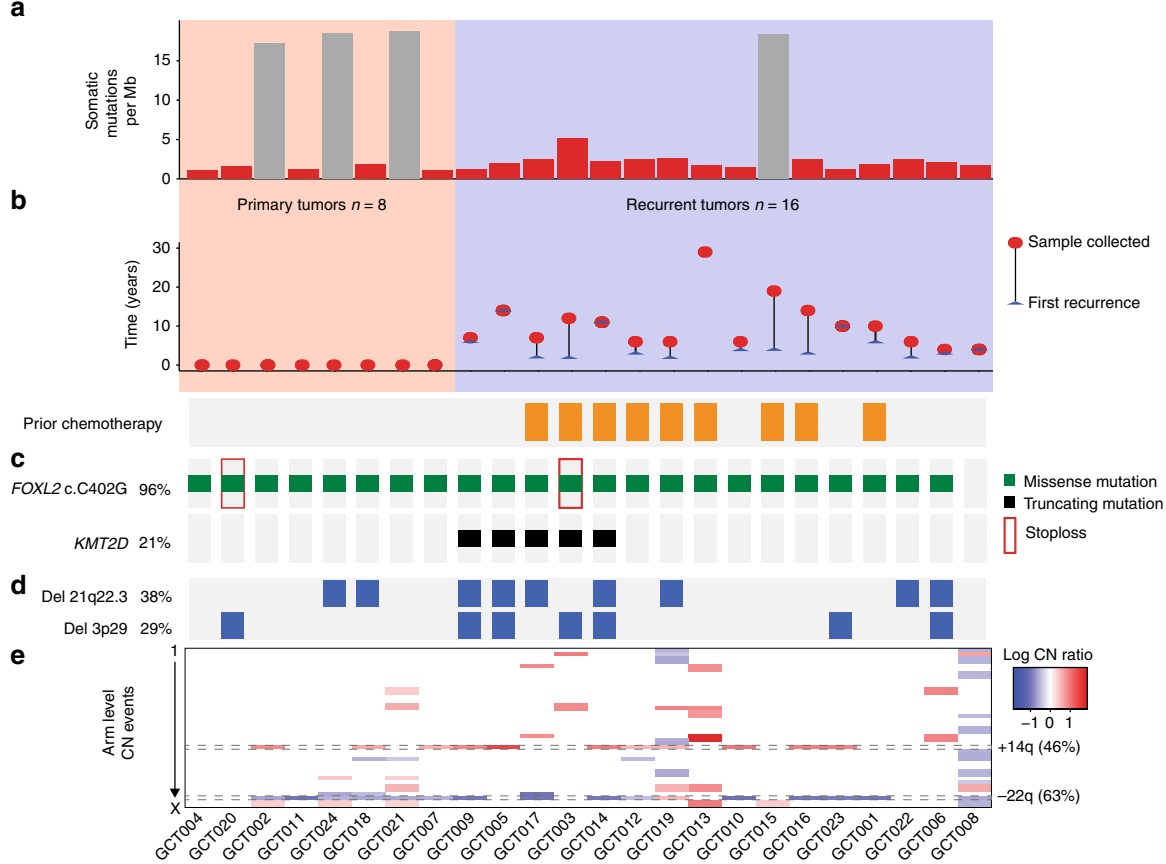

**Fig. 1** Mutational landscape of primary and recurrent aGCTs. **a** Somatic mutation rate per megabase (Mb) for primary ($n = 8$) and recurrent ($n = 16$) aGCTs. Gray bars indicate samples with no matched normal tissue, for which the mutation rate may be inflated by the inclusion of rare SNPs. **b** Time in years between diagnosis, first recurrence (blue arrow), and sample collection (red circle). The receipt of cytotoxic chemotherapy at any time between diagnosis and sample collection is indicated by an orange box. **c** Significantly mutated genes in aGCT. **d** Significantly recurrent focal CNVs. **e** Recurrent arm-level CNVs. Heat map indicates log CN ratio for CNV event. Mb: megabases, CN: copy number

| **Table 1 Somatic _KMT2D_-truncating mutations detected in sequenced aGCT cohorts** | | | | | | |
|---|---|---|---|---|---|---|
| **Sample** | **Cohort** | **Platform** | **_FOXL2_ c.C402G hotspot** | **Tumor type** | **_KMT2D_-truncating mutations** | |
| | | | | | **Nucleotide change** | **Protein change** |
| GCT003 | Exploratory | WES | Detected | Recurrent | c.7850dupC | p.Pro2617fs |
| GCT005 | Exploratory | WES | Detected | Recurrent | c.3299dupA | p.Asp1100fs |
| GCT009 | Exploratory | WES | Detected | Recurrent | c.15059delT | p.Lys5020fs |
| GCT014 | Exploratory | WES | Detected | Recurrent | c.11958_11959del | p.Ser3986fs |
| GCT017 | Exploratory | WES | Detected | Recurrent | c.9416_9419del | p.Pro3139fs |
| KGN | Cell line | DAS | Detected | Recurrent[18] | c.4085delA | p.Asn1362fs |
| GCT047 | Validation | Panel Sequencing | Detected | Recurrent | c.2277_2281del | p.Pro759fs |
| GCT062 | Validation | Panel Sequencing | Detected | Recurrent | c.2993delC | p.Pro998fs |
| GCT069 | Validation | Panel Sequencing | Detected | Recurrent | c.4418 + 1G > A | Splicing |
| GCT071 | Validation | Panel Sequencing | Detected | Primary | c.529_547del | p.Arg177fs |
| GCT072 | Validation | Panel Sequencing | Detected | Recurrent | c.C12715T | p.Gln4239X |
| GCT083 | Validation | Panel Sequencing | Detected | Recurrent | c.G2593T | p.Glu865X |
| WES: whole-exome sequencing, DAS: deep amplicon sequencing, FFPE: formalin-fixed, paraffin-embedded | | | | | | |

the combined aGCT cohort as a whole ($n = 79$ tumors; Fig. 2b), we found that _KMT2D_-truncating mutations exhibited a statistically significant enrichment in recurrent (10/43, 23%) compared to primary aGCTs (1/32, 3%; $p = 0.02$, two-sided Fisher's exact test).

**KMT2D loss detected by IHC.** In order to establish the relationship between _KMT2D_ mutation status and protein expression in aGCTs, we used IHC to detect KMT2D expression in tumor samples from the combined cohort (Supplementary Data 6). KMT2D signal was then analyzed quantitatively using multispectral imaging (see Methods). Although the majority of aGCTs had robust nuclear KMT2D expression, we identified a subset of tumors that exhibited loss of detectable nuclear KMT2D expression (Fig. 3a). When examining individual nuclei across all tumors in the combined cohort (Fig. 3b), nuclear KMT2D in aGCTs with truncating _KMT2D_ mutations ($n = 1,964,133$ nuclei)

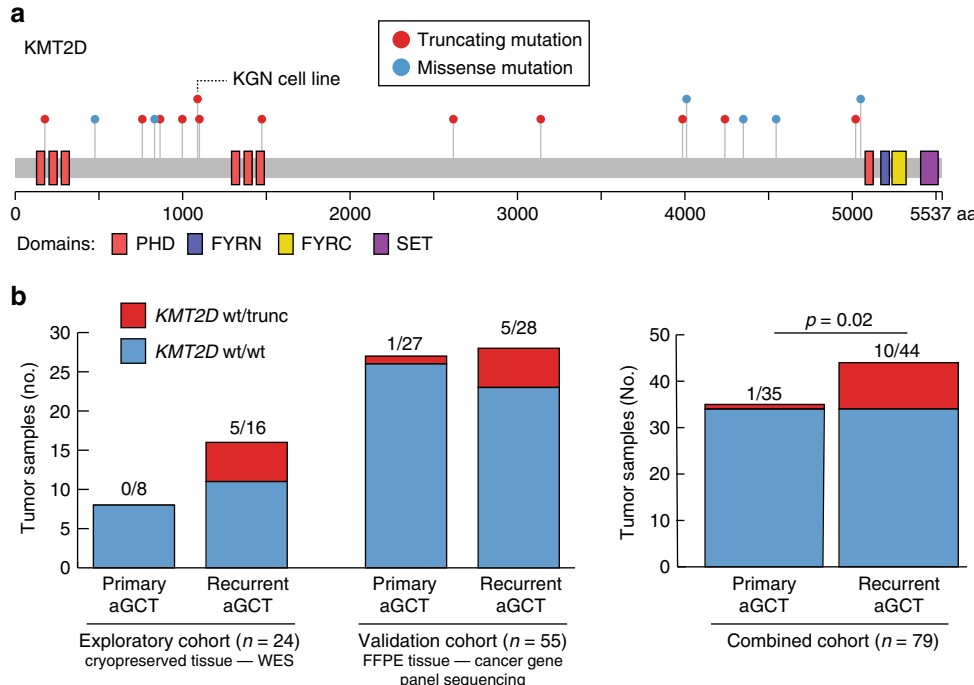

**Fig. 2** Truncating *KMT2D* mutations are associated with aGCT recurrence. **a** Overview of all somatic *KMT2D* mutations (non-synonymous SNVs + indels) detected in the combined cohorts. A frameshift indel detected in the KGN cell line is also indicated. **b** Rate of truncating *KMT2D* mutations identified in exploratory cryopreserved tissue cohort (*n* = 24) and validation FFPE tissue cohort (*n* = 55). *P* value from two-sided Fisher's exact test. WES: whole-exome sequencing, FFPE: formalin-fixed paraffin-embedded

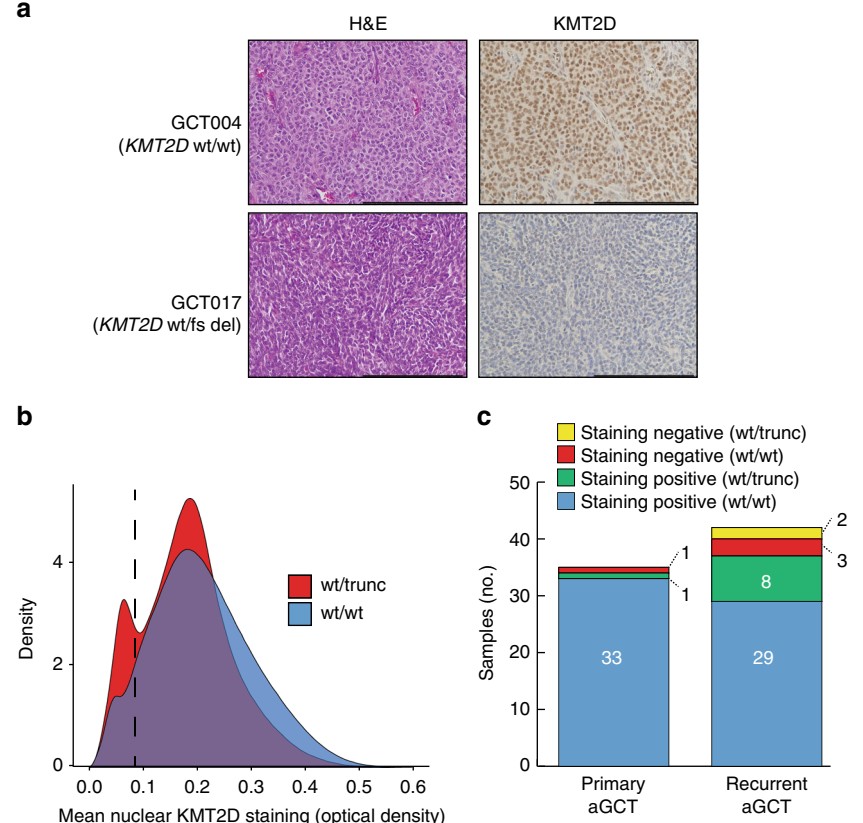

**Fig. 3** Nuclear KMT2D expression in primary and recurrent aGCTs. **a** Representative stained sections from tumors with positive (GCT004) and negative (GCT017) nuclear KMT2D expression. Scale bar = 200 μm. **b** Density plot of per nucleus mean KMT2D staining by KMT2D mutation status. Plots were derived from measurements made from individual nuclei in KMT2D wt/trunc (*n* = 1,964,133 nuclei) and KMT2D wt/wt (*n* = 12,121,768) tumors. Dotted line indicates numerical cutoff to distinguish "negative" from "positive" nuclei (see Methods). **c** Relationship between mono-allelic *KMT2D*-truncating mutation and loss of KMT2D protein expression by tumor type. fs: frameshift mutation, trunc: truncating mutation (any class)

exhibited a bi-modal distribution with a substantial subpopulation of KMT2D-negative nuclei. As a result, the distribution of measured nuclear KMT2D optical density in aGCTs with truncating KMT2D mutations exhibited a statistically significant difference ($p < 10^{-15}$, two-sided Wilcoxon's rank-sum test) when compared to the distribution in aGCTs without truncating KMT2D mutations ($n = 12,121,768$ nuclei). We next analyzed tumors as a whole by categorizing each tumor as "positive" or "negative" for nuclear KMT2D using stringent criteria (see Methods). Although loss of KMT2D protein expression was more frequent in recurrent (5/42, 12%) compared to primary (1/35, 3%) aGCTs, this difference was not statistically significant ($p = 0.21$, two-sided Fisher's exact test). Overall aGCTs exhibited a complex and partially overlapping pattern of mono-allelic KMT2D-truncating mutation and loss of KMT2D protein expression (Fig. 3c).

## Discussion

Here we report the genomic analysis of 79 primary and recurrent aGCTs, a rare gynecologic malignancy which remains difficult to treat in the recurrent setting. The tumors in these cohorts recapitulated previously described molecular characteristics of aGCTs, including both a high rate of FOXL2 c.C402G mutation[8] and a pattern of cytogenetic abnormalities notable for frequent arm-level gain of 14q and loss of 22q[21]. Thus, the aGCT samples analyzed in these cohorts are genetically representative of this rare tumor type. Along with these previously described molecular features of aGCTs, we also identified KMT2D-truncating mutations as a recurrent event in this tumor type.

We found significant differences in the genomic landscape of primary and recurrent aGCTs, including a higher mutation rate (SNVs + indels) in recurrent tumors. The observed difference in somatic mutation frequency between primary and recurrent aGCTs may reflect a combination of factors, including the time interval between diagnosis and sample collection, as well as prior exposure to cytotoxic chemotherapy. The most notable genetic distinction between primary and recurrent aGCTs was the significantly higher prevalence of mono-allelic KMT2D-truncating mutation in recurrent compared to primary tumors.

KMT2D is a tumor suppressor gene with histone lysine methyltransferase activity that is the target of frequent inactivating mutations in several tumor types, including medulloblastoma, diffuse large B cell lymphoma (DLBCL), and follicular lymphoma[22,23]. The KMT2D protein contains a conserved C-terminal SET (Su(var)3–9, Enhancer-of-zeste, and Trithorax) domain that imparts catalytic histone lysine methyltransferase activity. KMT2D is thought to regulate transcription factor access to distal enhancers through the placement of histone methylation marks at these elements[24,25]. The key role played by FOXL2 in granulosa cell development and aGCT formation has led to the speculation[26] that this protein may function as a pioneering transcription factor with chromatin-remodeling actions similar to those observed among other forkhead family transcription factors[27]. The co-occurrence of FOXL2 c.C402G hotspot mutation[8] and KMT2D inactivation in aGCT suggests that these lesions may function cooperatively, possibly through the regulation of distal enhancer elements important for the gene expression programs associated with aGCT. In addition to truncating KMT2D mutations, we also identified mutations resulting in the loss of the FOXL2 stop codon as a rare recurrent mutation in aGCT (3/79 tumors, 4%). This mutation is predicted to increase the length of the FOXL2 protein by >40%, but additional studies will be necessary to determine the oncogenic function, if any, of this unusual mutation.

A prior analysis of KMT2D inactivation in DLBCL identified a subset of such tumors with loss of KMT2D protein expression despite the absence of detectable mono-allelic or bi-allelic inactivating mutations[20]. This observation suggests that exome sequencing may not identify all tumors with functional KMT2D inactivation, which may also occur via epigenetic mechanisms or by genetic mechanisms not captured by exome sequencing platforms using short paired-end reads (e.g., large indels or copy number neutral structural variation). Consistent with these observations in DLBCL, we also identified a subset of aGCTs that lack KMT2D staining by IHC but nevertheless do not have detectable KMT2D-truncating mutations. We did not observe a high degree of overlap between mono-allelic KMT2D truncation and loss of protein expression, and it remains unknown whether these states represent similar disease phenotypes. The functional relationship in aGCT between mono-allelic KMT2D truncation and loss of protein expression requires exploration in future studies.

Several clinical factors including initial tumor stage[3,4], incomplete staging at the time of initial surgery[5], and intraoperative tumor rupture[6] have been reported to be associated with the risk of aGCT recurrence. Given the rarity of KMT2D inactivation among primary aGCTs, a much larger series will be required to formally evaluate KMT2D inactivation as an independent risk factor for recurrence. Of the 15 patients in the combined cohort found to have aGCTs with either mono-allelic KMT2D inactivation or loss of KMT2D protein expression, data on initial stage was available for 13 patients. It is interesting to note that 12 out of 13 (92%) of these patients had stage I disease at the time of diagnosis, a rate consistent with the overall stage distribution of newly diagnosed aGCT. Additional genomic studies of paired primary and recurrent tumor tissue will be needed to establish whether all instances of KMT2D inactivation in aGCT are detectable at the time of diagnosis or if such lesions can occur stochastically in a subset of tumors during post-treatment surveillance, portending eventual relapse.

## Methods

**Patients and samples.** This retrospective study was approved by the institutional review board at The University of Texas MD Anderson Cancer Center (protocols PA16-0891 and LAB02-188) and patients provided written informed consent for tissue storage and analysis. All tumors included in the "exploratory" and "validation" cohorts had pathologic diagnoses of aGCT rendered by a pathologist at The University of Texas MD Anderson Cancer Center. For the "exploratory" cohort, cryopreserved tissue samples were obtained from The University of Texas MD Anderson Cancer Center Multidisciplinary Gynecologic Cancer Tumor Bank, and matched fresh-frozen normal tissue or peripheral blood corresponding to 20 of these tumors was also obtained. Tumors were excluded if the initial surgical pathology report indicated the presence of >10% of non-aGCT histology. For DNA isolation from banked samples in the "exploratory" cohort, frozen tissue samples were cryosectioned and a section adjacent to the portion submitted for sequencing underwent hematoxylin and eosin staining to assess tumor cell content. These adjacent sections were reviewed by a gynecologic pathologist (R.R.B.) and only tumor samples with an estimated tumor cell content >60% without significant necrosis were included in the "exploratory" cohort.

For the independent "validation" cohort, sections prepared from FFPE tumor blocks were reviewed for tumor cell content and included if the estimated tumor cell content was >60% without significant necrosis. A total of 55 tumor FFPE blocks met the criteria for inclusion. After pathologic review to ensure the absence of visible tumor cells (R.R.B.) we were able to identify normal tissue FFPE blocks corresponding to 44 of these tumors. Clinical data was abstracted from the medical record and managed using REDCap[28] (Research Electronic Data Capture) electronic data capture tools hosted at The University of Texas MD Anderson Cancer Center.

**Whole-exome sequencing.** Following extraction from frozen tissue, DNA samples were submitted for 76 bp short-read WES on Illumina HiSeq 2000 (Illumina) after SureSelect Human All Exon V4 library preparation (Agilent Technologies). The target coverage was 200× for tumor samples and 100× for matched normals. For each sample, the reads were mapped to the hg19 reference genome using BWA-MEM[29] and processed using the Genome Analysis Toolkit best practices[30],

followed by somatic variant calling using matched normals (MuTect[31], Pindel[32]). For samples in the exploratory cohort without a matched normal, a bespoke "common normal" BAM file was used which comprised of down-sampled paired-end WES reads derived from peripheral blood from five donors. Aside from the use of this "common normal" BAM file, all other aspects of the mutation calling pipeline were identical for samples with and without a matched normal. High-quality SNVs were defined as those with a minimum tumor read depth of ≥20, minimum matched normal read depth of ≥10, and minimum alternate allele frequencies in the tumor and normal as ≥0.02 and ≤0.02, respectively. Small insertions and deletion calls were removed if within 25 bp of a repetitive element. Tumor samples were included in the "exploratory" cohort only if the estimated tumor purity from B allele frequencies was >0.15 (Sequenza).[33] For calculation of mutation rate, the denominator was determined on a per tumor basis using the length of genomic regions meeting criteria for mutation calling (minimum tumor read depth of ≥20 and minimum matched normal read depth of ≥10).

**Mutation validation**. The five somatic *KMT2D* variants identified in the WES exploratory cohort were all verified using a custom-designed TruSeq Custom Amplicon v1.5 panel (Illumina) following the manufacturer's instructions. The regions of *KMT2D* and *FOXL2* interrogated by the TruSeq panel are included as Supplementary Data 4. The *FOXL2* stoploss indels were also verified in a similar manner. For this procedure, genomic DNA was isothermally amplified using custom amplicon primers, followed by a PCR step resulting in the addition of sequencing adapters and indices. PCR products were cleaned up using Agencourt AMPure XP beads (Beckman Coulter) and normalized using a bead-based normalization procedure (Illumina). Normalized amplicon sequencing libraries were then pooled and sequenced on an Illumina MiSeq (Illumina) with a target coverage of 1000×.

**Cancer gene panel sequencing**. For tumor and normal FFPE samples in the independent "validation" cohort, 10–15 scrolls of 10 μm thickness were prepared from the selected blocks and DNA was extracted using the QIAamp DNA FFPE Tissue Kit (Qiagen). Tumor and normal samples were submitted for 76 bp short-read paired-end sequencing on Illumina HiSeq 2000 (Illumina) after library preparation and capture using the T200.1 panel[19], a hybrid capture platform targeting 201 cancer-related genes that has been previously validated for use with FFPE samples. Read alignment and somatic variant calling were performed as described above for WES, with the addition of stringent filtering criteria applied to indels. Specifically, a minimum tumor and normal coverage of 50× was required at the site of the candidate indel. In addition, the tumor fraction for the candidate indel was required to be >0.1 with no reads in the corresponding normal sample showing the indel.

**Copy number analysis from WES data**. Copy number variation (CNV) data were derived from WES data using a bespoke R package, as previously described[34]. Briefly, reads were counted in each exon region using bedtools and the log 2 ratio of tumor versus normal reads were then calculated for each exonic region after adjustment for the total mapped reads in that region. The log 2 copy number ratios were then segmented using the DNAcopy package of Bioconductor[35]. Recurrent arm-level and focal CNVs were then identified with GISTIC2.0[36], using a log 2 CN ratio threshold of <−0.3 or >0.3 for deletions and amplifications, respectively, with a 95% confidence threshold. A CNV was determined to be arm level if it accounted for at least 70% of segments for a particular chromosomal arm. *P* values for recurrent arm-level and focal CNVs were calculated by GISTIC2.0 for each segment, followed by correction using the Benjamini–Hochberg false discovery rate method to produce the adjusted *p* values (*q* values) reported in the text. A *q* value threshold of <0.05 was used.

In order to reduce the risk of germline CNVs being identified as significantly recurrent somatic events, the four tumors in the "exploratory" cohort without a matched normal were excluded from the GISTIC2.0 analysis of focal CNVs. For the two regions identified as significantly recurrent in the "exploratory" cohort, the CN profiles for the four tumors without a matched normal were then reviewed manually to determine if these CNVs were present. Due to the use of peripheral blood as a matched normal tissue, we excluded as spurious significant focal CNVs identified by GISTIC2.0 that included T cell receptor genes (*TCRA*, *TCRG*, *TCRB*) as these were likely due to rearrangement of these loci in T cell lymphocytes.

**Identification of significantly mutated genes**. GenomeMuSiC2[13] v.0.4 was used to identify significantly mutated genes from among the set of SNVs and indels identified in the "exploratory" WES cohort. In order to reduce the risk of rare SNPs being identified as significantly recurrent mutations, the four tumors in the "exploratory" cohort without a matched normal were excluded from this analysis. The GenomeMuSiC2 statistical model incorporates gene size, sequencing coverage depth, and background mutation rate while provided for rigorous control of the false discovery rate. For detecting significantly mutated genes, a minimum tumor coverage of 8 and a minimum normal coverage of 6 was required. Primary and recurrent tumors were analyzed together and a false discovery rate of <0.1 was used.

**Cell culture**. The KGN granulosa cell tumor cell line[15,16] was obtained from the RIKEN BioResource Center (RCB1154) and was maintained in antibiotic-free 1:1 Dulbecco's modified Eagle's medium/HamF-12 growth medium (HyClone) with 10% fetal bovine serum added. The identity of this cell line was independently confirmed using short tandem repeat testing, and the line was regularly tested to exclude the presence of mycoplasma. A *KMT2D* frameshift indel (c.4085delA, p. Asn1362fs) previously cataloged by the Cancer Cell Line Encyclopedia[17] and the COSMIC Cell Lines Project[18] was confirmed by deep amplicon sequencing using a TruSeq v1.5 Custom Amplicon panel as described above (Illumina).

**Immunohistochemical analysis of KMT2D expression**. IHC analysis was performed on 4 μm FFPE tumor sections prepared from blocks corresponding to the same "exploratory" and "validation" cohort samples submitted for WES and cancer gene panel sequencing, respectively. Of the 79 aGCTs analyzed by genomic sequencing (WES or T200.1 cancer gene panel sequencing), one sample from the "exploratory" cohort did not have FFPE material available and one sample from the "validation" cohort had only a poor quality FFPE block that was not suitable for IHC; thus, 77 out of the 79 tumors in the combined cohort were able to be analyzed by IHC.

All slides were de-paraffinized in xylene and rehydrated. A heat-induced antigen retrieval step was employed using a citrate buffer at pH 6.0 (LabVision). Slides were incubated in peroxidase blocking solution (Dako) for 10 min at room temperature. All antibodies were then applied for 60 min at room temperature after dilution in an antibody diluent. KMT2D antibody was diluted in serum-free diluent (Dako). Slides were washed, and then incubated with anti-rabbit HRP-labeled polymer (Dako). Staining was visualized with diaminobenzidine (DAB) reagent (LabVision) and slides were counterstained with hematoxylin to visualize nuclei. KMT2D expression was detected using a previously validated[20] rabbit polyclonal antibody directed against the KMT2D C terminus (HPA035977, Sigma-Aldrich) (1:500 dilution).

**Automated IHC image analysis**. A Vectra 3.0 Automated Quantitative Pathology Imaging System (Perkin-Elmer) was used to perform multispectral analysis of stained tumor sections. Representative FFPE sections with single stains (DAB or hematoxylin) were used to construct a spectral library for use in multispectral image analysis. For each stained section, the Vectra 3.0 system was then used to collect multispectral imaging across a designated set of non-overlapping high-power fields (HPFs) designed to cover between 10 and 50% of total tumor area on a slide. InForm analysis software (Perkin-Elmer) was used to train a tissue classifier, which was then used to segment areas of tumor tissue present on each section. After segmentation of tumor tissue, the InForm software was then used to segment nuclei. The optical density of hematoxylin and DAB was then determined for each pixel within the segmented nuclei.

As a quality control measure to remove HPFs with aberrant segmentation, individual HPFs were filtered if either the median nuclear area, median nuclear axis ratio, or median nuclear hematoxylin was determined to be a statistical outlier among the dataset as a whole (defined as more than 1.5 IQRs below the first quartile or above the third quartile). Of 6793 HPFs segmented, 433 were filtered using these criteria resulting in 6360 (94%) HPFs used for subsequent analyses.

In order to establish a stringent definition of the optical density associated with "negative" nuclear KMT2D expression, a section from a representative tumor (GCT004) was stained with hematoxylin alone (i.e., no DAB) and analyzed in parallel to the KMT2D-stained sections. A tumor sample was classified as "KMT2D negative" if the median nuclear optical density of the DAB signal was within the range of nuclear optical density DAB signal for the hematoxylin-only section (specifically <0.09 optical density). Using this approach, 6/77 (8%) of aGCTs in the combined cohort were classified as "KMT2D negative."

**Statistical analyses**. Categorical comparisons were performed using a Fisher's exact test and comparisons between continuous variables were done using a Wilcoxon's rank-sum test. All statistical comparisons were two sided and a *p* value <0.05 was considered significant. Statistical tests were performed using R version 3.3.1.

**Data availability**. The whole-exome sequencing and T200.1 cancer gene panel sequencing data related to this study have been deposited with the European Genome-phenome Archive (EGA) under access code EGAS00001002833.

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

## Acknowledgements

We would like to acknowledge Dr. Jared Burks (Co-Director of the Flow Cytometry and Cellular Imaging Core Facility) for his expert assistance with multispectral imaging and quantitation systems. This work was supported by a Judy's Mission Ovarian Cancer Foundation grant (to R.T.H.), a Cancer Prevention and Research Institute of Texas (CIPRIT) grant (R1205; to P.A.F.), a T32 training grant for gynecologic oncology (CA101642; to K.H.L.), and by The MD Anderson Cancer Center Support Grant (CA016672) that supports the Flow Cytometry and Cellular Imaging Facility and Sequencing and Microarray Core Facility. R.R.B. and The University of Texas MD Anderson Cancer Center Multidisciplinary Gynecologic Cancer Tumor Bank are supported in part by NIH P50CA83639 SPORE in Ovarian Cancer.

## Author contributions

R.T.H. and P.A.F. conceived of the initial experimental design and wrote the manuscript. R.T.H., J.C., C.T., H.C.B., C.G., L.L., T.N., R.T., S.T., and J.Z. carried out experiments and analyzed data. K.H.L., D.M.G., K.R., R.R.B., and P.A.F. supervised the study and assisted with experimental design. All authors reviewed and revised the manuscript.

## Additional information

**Competing interests:** The authors declare no competing interests.

