## [Peer Review File · Nature Communications]

Reviewers' comments:

Reviewer #1 (Remarks to the Author):

Hillman and colleagues assess the genomic landscape of primary and recurrent adult-type granulosa cell tumours (aGCTs) using exome and targeted sequencing approaches. They identified mutations and loss of expression of KMT2D in recurrent aGCTs, suggesting convergence on loss of KMT2D as a driver of disease recurrence. This is an interesting and well executed study, that analyses a large number of samples for a rare cancer type.

A few minor comments:

- 1) Please discuss the findings of Pilsworth et al (Modern Pathology (2018)) who performed whole genome sequencing on aGCTs in the introduction.
- 2) In the Methods section please clarify (a) how the mutation rate was calculated for cases without a matched normal, (b) details of the TruSeq Amplicon panel such as which regions of the genes were targeted, and (c) the data availability of the targeted sequencing.
- 3) Does the allele frequency of the KMT2D mutations suggest that the mutations are clonal or subclonal?
- 4) Line 195 states the "discovery" cohort, while lines 200 and 205 describe an "exploratory" cohort - please clarify if these are the same cohort or different cohorts.

Reviewer #2 (Remarks to the Author):

The author's demonstrate a recurrent genetic event (KMT2D truncating mutation) that it more commonly seen in recurrent aGCT, compared to primary tumors. Loss of expression of KMT2D protein (as assessed by immunohistochemistry) is also more common in recurrent aGCT, compared to primary aGCT. Loss of expression correlates imperfectly with mutation status, in that most tumors with a truncating KMT2D mutation have detectable protein, while most tumors with loss of expression do not have a mutation, when a simple positive versus negative classification of the immunostaining is used.

1. Loss of KMT2D protein expression is not significantly more common in recurrent aGCT compared to primary aGCT (1/35 vs 5/42). The conclusion that either mutation (mono-allelic inactivation) or loss of expression is more frequent in recurrent than primary aGCT is not supported by the data. They have combined tumors with either loss of expression or mutation; while the tumors with mutation are more common in the recurrent aGCT those with loss of expression are not, and combining them is not justified, in my opinion, given the very imperfect correlation between expression and mutation.
2. In their concluding comments, the authors note "an association between KMT2D inactivation and disease recurrence" and go on to comment that "the lack of complete follow up data for patients with primary included in these cohorts precludes a formal evaluation of KMT2D inactivation as an independent risk factor for recurrence". I worry that this wording could be misinterpreted by readers as indicating that either the presence of KMT2D mutation or loss of expression is of prognostic significance, when they are not. To suggest that multivariable analysis to examine independent prognostic significance is worth doing is not supported by the data, as only a single primary aGCT has a mutation and one show loss of protein expression (a tumor that lacks a mutation). With only 2/35 primary aGCT showing "inactivation" it is not realistic to talk of multivariable analysis.
3. It is unfortunate that there are no paired primary and recurrent samples in the study. Even a single such sample set could have proved informative in addressing the question of whether the mutation is detectable in the primary tumor, and if so at what frequency. To go back and do deep sequencing on multiple samples from the primary tumor in those cases where a mutation had been detected in the recurrence would have been of great interest.
4. I don't understand the last sentence of the main text: "Additional prospective studies will be

needed to establish whether all instances of KMT2D inactivation in aGCT are detectable at the time of diagnosis or if such lesions can occur stochastically in a subset of tumors during post-treatment surveillance, portending eventual relapse". As noted above, I believe that the ability to detect mutations in tumors that have mutations at the time of recurrence will be most directly addressed retrospectively, with thorough examination of paired primary tumor samples and I don't see how a prospective approach has merit.

RESPONSE TO REVIEWERS

Reviewers' comments:

Reviewer #1 (Remarks to the Author):

Hillman and colleagues assess the genomic landscape of primary and recurrent adult-type granulosa cell tumours (aGCTs) using exome and targeted sequencing approaches. They identified mutations and loss of expression of *KMT2D* in recurrent aGCTs, suggesting convergence on loss of *KMT2D* as a driver of disease recurrence. This is an interesting and well executed study, that analyses a large number of samples for a rare cancer type.

A few minor comments:

1) Please discuss the findings of Pilsworth et al (Modern Pathology (2018)) who performed whole genome sequencing on aGCTs in the introduction.

The important work by Pilsworth et al describing recurrent *TERT* C228T promoter mutations in adult granulosa cell tumors was published contemporaneously with our initial manuscript submission, and thus was not available at that time for citation in our manuscript. We have now included a discussion of these data in our introduction, and cite the published work [see line 56].

2) In the Methods section please clarify (a) how the mutation rate was calculated for cases without a matched normal, (b) details of the TruSeq Amplicon panel such as which regions of the genes were targeted, and (c) the data availability of the targeted sequencing.

- (a) The reviewer raises an important point, since the examination of whole exome tumor sequencing data without the use of a matched normal sample raises unique analytical issues. In the methods section, we discuss how a bespoke “common normal” BAM file was constructed from down-sampled paired-end whole exome sequencing reads derived from the peripheral blood of 5 donors. For samples without a matched normal, we used this “common normal” to call single nucleotide variants and small insertion/deletions using the same pipeline used for the other samples. We have now added text to the methods section clarifying this point [see line 242].
- (b) We have now included the amplicon regions for *KMT2D* and *FOXL2* that were covered by the TruSeq Custom Amplicon v1.5 Panel (Illumina) as a new Supplementary Table 4.

(c) We have amended the Data Availability section of the manuscript to explicitly state that the T200.1 cancer gene panel sequencing data are available along with the whole exome sequencing data through the European Genome-phenome Archive (EGA) under access code EGAS00001002833 [see line 361].

3) Does the allele frequency of the *KMT2D* mutations suggest that the mutations are clonal or subclonal?

For each of the 5 *KMT2D* indels observed in the exploratory cohort, the raw variant allele frequency, estimated tumor purity calculated using Sequenza, and normalized variant allele frequency (raw VAF/tumor purity) are shown in the following table. Data in this table is derived from that presented in the manuscript within Supplementary Table 2 and Supplementary Table 3, but is presented here in order to address the point raised by the Reviewer.

Sample Name	KMT2D Nucleotide Change	Raw VAF	Tumor Purity	Normalized VAF
GCT003_DNA_RecurrentTumor	c.7850dupC	0.28	0.94	0.30
GCT005_DNA_RecurrentTumor	c.3299dupA	0.05	0.17	0.29
GCT009_DNA_RecurrentTumor	c.15059delT	0.33	0.92	0.36
GCT014_DNA_RecurrentTumor	c.11958_11959del	0.40	0.46	0.88
GCT017_DNA_RecurrentTumor	c.9416_9419del	0.34	0.97	0.35

*VAF = variant allele frequency

Notably, we did not detect focal or broad copy number variants encompassing *KMT2D* in any of the tumors containing truncating *KMT2D* indels. As can be seen in the table, the normalized variant allele frequency for 4 of the 5 *KMT2D* indels falls in the range 0.29-0.36. For variant allele frequencies in this range, the distinction between clonal and sub-clonal mutations is ambiguous and thus the manuscript does not contain a conclusive statement in this regard. We have added text to the discussion section to specifically address the question of *KMT2D* indel clonality [see line 93].

4) Line 195 states the "discovery" cohort, while lines 200 and 205 describe an "exploratory" cohort - please clarify if these are the same cohort or different cohorts.

In the manuscript, the terms "discovery cohort" and "exploratory cohort" referred to the same initial set of 24 cryopreserved adult type granulosa cell tumors analyzed by whole exome sequencing. We have now standardized all references to this cohort to refer only to the "exploratory cohort", in concordance with Figure 2B.

Reviewer #2 (Remarks to the Author):

The author's demonstrate a recurrent genetic event (*KMT2D* truncating mutation) that it more commonly seen in recurrent aGCT, compared to primary tumors. Loss of expression of *KMT2D* protein (as assessed by immunohistochemistry) is also more common in recurrent aGCT, compared to primary aGCT. Loss of expression correlates imperfectly with mutation status, in that most tumors with a truncating *KMT2D* mutation have detectable protein, while most tumors with loss of expression do not have a mutation, when a simple positive versus negative classification of the immunostaining is used.

1. Loss of *KMT2D* protein expression is not significantly more common in recurrent aGCT compared to

primary aGCT (1/35 vs 5/42). The conclusion that either mutation (mono-allelic inactivation) or loss of expression is more frequent in recurrent than primary aGCT is not supported by the data. They have combined tumors with either loss of expression or mutation; while the tumors with mutation are more common in the recurrent aGCT those with loss of expression are not, and combining them is not justified, in my opinion, given the very imperfect correlation between expression and mutation.

We agree with the reviewer that the relationship between *KMT2D* mono-allelic truncating mutation and loss of detectable *KMT2D* protein expression is admittedly complex. This has been elegantly demonstrated in prior studies of *KMT2D* mutation in diffuse large B cell lymphoma (DLBCL), published by the laboratory of Laura Pasqualucci [see Zhang et al “Disruption of *KMT2D* perturbs germinal center B cell development and promotes lymphomagenesis” *Nat Med.* 2015 Oct;21(10)]. In Figure 1D-E of this paper, the authors provide a comprehensive overview of the observed rates of *KMT2D* mutation and their relationship to loss of protein expression. In the case of DLBCL, loss of *KMT2D* protein expression was observed in approximately 11% (2/18 cases) of tumors with mono-allelic *KMT2D* truncating mutations. The authors observed loss of *KMT2D* protein expression in 12% (9/77 cases) of *KMT2D* wild type tumors, which they hypothesized may be due to epigenetic mechanisms. We agree with the Reviewer that given the complex relationship between mono-allelic *KMT2D* truncating mutation and loss of *KMT2D* protein expression, these categories should not be combined for the purpose of data presentation or statistical tests.

We have therefore made the following changes to the manuscript in order to address the important points raised by the Reviewer:

- We no longer combine the rate of mono-allelic *KMT2D* mutation and the rate of loss of *KMT2D* protein expression for the purpose of statistical comparison. We have removed reference to these statistical tests from the abstract and manuscript text.
- Instead, the emphasis in the abstract and text is now placed on the statistically significant difference in the rate of mono-allelic *KMT2D* truncating mutation between primary and recurrent tumors.
- We have modified Figure 3C in the manuscript so as to no longer combine mono-allelic *KMT2D* mutation and the rate of loss of *KMT2D* protein expression. This figure now functions merely as a concise summary of the complex relationship between mono-allelic *KMT2D* truncating mutation and loss of *KMT2D* protein expression, in a manner analogous to the presentation of similar data for DLBCL reported by Zhang, et al, as published in *Nature Medicine*.
- We have expanded the Discussion section to explicitly address the low degree of overlap we observe between mono-allelic *KMT2D* truncating mutation and loss of *KMT2D* protein expression [see line 191].

2. In their concluding comments, the authors note "an association between *KMT2D* inactivation and disease recurrence" and go on to comment that "the lack of complete follow up data for patients with primary included in these cohorts precludes a formal evaluation of *KMT2D* inactivation as an independent risk factor for recurrence". I worry that this wording could be misinterpreted by readers as indicating that either the presence of *KMT2D* mutation or loss of expression is of prognostic significance, when they are not. To suggest that multivariable analysis to examine independent prognostic significance is worth doing is not supported by the data, as only a single primary aGCT has a mutation and one show loss of protein expression (a tumor that lacks a mutation). With only 2/35 primary aGCT showing "inactivation" it is not realistic to talk of multivariable analysis.

We thank the reviewer for pointing out the imprecision in the manuscript text where we discuss the possible investigation of *KMT2D* inactivation as a risk factor for recurrence. We agree that even with complete follow up data (which is not available for our cohort), such an analysis would not be possible given the rarity of *KMT2D* inactivation in primary tumors (2/35 cases). Multivariable analysis to examine the independent prognostic significance of *KMT2D* inactivation remains an interesting question, but this would require a much larger series of primary tumors. We have amended the text to clarify these points in a manner that is consistent with the Reviewer's comment [see line 198].

3. It is unfortunate that there are no paired primary and recurrent samples in the study. Even a single such sample set could have proved informative in addressing the question of whether the mutation is detectable in the primary tumor, and if so at what frequency. To go back and do deep sequencing on multiple samples from the primary tumor in those cases where a mutation had been detected in the recurrence would have been of great interest.

We wholeheartedly agree with the reviewer that the analysis of paired primary/recurrent samples would be of great interest. Unfortunately, we do not have paired archived tissue of appropriate quality for any adult type granulosa cell tumors. If such paired samples were available then we would absolutely have analyzed them as part of this study. This circumstance is likely due to the nature of M.D. Anderson Cancer Center as a cancer care referral center. Since the large majority of patients with adult type granulosa cell tumors are cured by surgery alone, when treated at our center these patients most often choose to undertake clinical surveillance follow-up closer to home. Conversely, patients seen for management of recurrent disease at our institution almost universally had a primary surgery performed by a local gynecologist many years prior.

4. I don't understand the last sentence of the main text: "Additional prospective studies will be needed to establish whether all instances of *KMT2D* inactivation in aGCT are detectable at the time of diagnosis or if such lesions can occur stochastically in a subset of tumors during post-treatment surveillance, portending eventual relapse". As noted above, I believe that the ability to detect mutations in tumors that have mutations at the time of recurrence will be most directly addressed retrospectively, with thorough examination of paired primary tumor samples and I don't see how a prospective approach has merit.

The reviewer raises an important point that requires clarification in the manuscript text. This point pertains to the question of whether *KMT2D* inactivating mutations are always detectable at the time of diagnosis. We agree with the Reviewer that this question will likely be answered most expeditiously using paired primary and recurrent tissue samples. We have amended the discussion section to reflect this clarification, and thank the reviewer for this insight [see line 204].

REVIEWERS' COMMENTS:

Reviewer #1 (Remarks to the Author):

Thank you for addressing all of the reviewers comments appropriately.

REVIEWERS' COMMENTS:

Reviewer #1 (Remarks to the Author):

Thank you for addressing all of the reviewers comments appropriately.

We appreciate the and acknowledge the time and thoughtful attention provided by the reviewers during both rounds of review.

NOTE: Reviewer #2 provided no remarks to the author.